# Side-Effects following Oxford/AstraZeneca COVID-19 Vaccine in Tororo District, Eastern Uganda: A Cross-Sectional Study

**DOI:** 10.3390/ijerph192215303

**Published:** 2022-11-19

**Authors:** Jagire Onyango, David Mukunya, Agnes Napyo, Ritah Nantale, Brian T. Makoko, Joseph K. B. Matovu, Benon Wanume, David Okia, Francis Okello, Sam Okware, Peter Olupot-Olupot, Yovani Lubaale

**Affiliations:** 1Department of Community and Public Health, Busitema University, Mbale P.O. Box 236, Uganda; 2Department of Research, Nikao Medical Center, Kampala P.O. Box 10005, Uganda; 3Department of Nursing, Busitema University, Mbale P.O. Box 236, Uganda; 4Department of Disease Control and Environmental Health, Makerere University School of Public Health, Kampala P.O. Box 7072, Uganda; 5Department of Research, Uganda National Health Research Organizations, Kampala P.O. Box 465, Uganda; 6Department of Research, Mbale Clinical Research Institute, Mbale P.O. Box 1966, Uganda

**Keywords:** side-effects, COVID-19, Oxford/AstraZeneca vaccine, Uganda

## Abstract

Effective, safe and proven vaccines would be the most effective strategy against the COVID-19 pandemic but have faced rollout challenges partly due to fear of potential side-effects. We assessed the prevalence, profiles, and predictors of Oxford/AstraZeneca vaccine side-effects in Tororo district of Eastern Uganda. We conducted telephone interviews with 2204 participants between October 2021 and January 2022. Multivariable logistic regression was conducted to assess factors associated with Oxford/AstraZeneca vaccine side-effects using Stata version 15.0. A total of 603/2204 (27.4%) of the participants experienced one or more side-effects (local, systemic, allergic, and other side-effects). Of these, 253/603 (42.0%) experienced local side-effects, 449/603 (74.5%) experienced systemic side-effects, 11/603 (1.8%) experienced allergic reactions, and 166/603 (27.5%) experienced other side-effects. Ten participants declined to receive the second dose because of side-effects they had experienced after the first dose. Previous infection with COVID-19 (adjusted odds ratio (AOR): 4.3, 95% confidence interval (95% CI): 2.7–7.0), being female (AOR: 1.3, 95% CI: 1.1–1.6) and being a security officer (AOR: 0.4, 95% CI: 0.2–0.6) were associated with side-effects to the Oxford/AstraZeneca vaccine. We recommend campaigns to disseminate correct information about potential side-effects of the Oxford/AstraZeneca vaccine and strengthen surveillance for adverse events following vaccination.

## 1. Introduction

Globally, countries are grappling with low coronavirus disease 2019 (COVID-19) vaccine acceptance, even though vaccines are known to save lives [1]. The hesitancy is partly caused by the fear of vaccine side-effects [2]. Furthermore, the hasty manner with which vaccines under emergency authorization, such as the COVID-19 vaccines, are approved resulted in unease among some members of the scientific community. The World Health Organization (WHO) estimates that immunization programs worldwide prevent 2–3 million deaths from vaccine-preventable diseases every year [3], and are not only cost-effective but an essential element of preventative healthcare. Vaccines work with our body’s natural defenses to build protection against diseases in a process called immunization. A study conducted in the United Kingdom compared infection rates among a subset of vaccinated individuals reported a significant buildup of immunity after 12 days following vaccination with Oxford/AstraZeneca [4]. This interaction and other aspects of vaccines may cause untoward experiences such as swelling, pain, redness at the injection site, fever, headache, dizziness, joint pain, fainting, nausea, vomiting, diarrhea, and a rash among the vaccine recipients.

Vaccines are also critical to the prevention and control of infectious disease outbreaks; therefore, an effective, safe, proven and widely acceptable COVID-19 vaccine would be a great tool for controlling the pandemic [5]. Immunization is one of the most cost-effective health investments with proven strategies that make it accessible to even the most hard-to-reach and vulnerable populations [6]. However, not only does a vaccine need to be safe and effective, it also must be accepted by those at greatest risk of harm from the disease [7]. COVID-19 vaccine acceptance by a large proportion of the population would also offer protection to the other people who remain unimmunized, a phenomenon called herd immunity. Reported serious side-effects, inconsistent information, conspiracy theories, and geopolitics seem to be the drivers of poor acceptance at this level.

Uganda initially acquired about 900,000 doses of Oxford/AstraZeneca vaccine (Covishield), manufactured by the serum institute of India, and embarked on the vaccination campaign in earnest. However, low vaccine acceptance and hesitancy in Uganda are common [8]. This could be attributed to fear of potential risks that can be encountered, primarily where a vaccine has not been well evaluated [8]. As of 30 April 2021, the vaccination coverage in Uganda was at 330,077/990,000 of the available doses, representing about 33% achievements for the country (MOH press statement on COVID-19 updates). However, by the same date, the Tororo district had posted well over 85.9% utilization, with 6865 doses of the available 8000 dispensed, showing a fairly good acceptance. A study conducted in Western Uganda concluded that government needs to prioritize vaccine acceptance strategies, especially among the risky groups in the community, to ensure a successful vaccination process [8]. The same study found that the level of vaccine acceptance (53.6%) and risk perception (46.7%) was relatively average in Western Uganda. High-risk groups such as health workers are targeted with this vaccine to ensure stability in the system in case an overwhelming epidemic threatens to derail service delivery. The surveillance system in place may not be relied upon to provide conclusive data on adverse events following immunization. Anecdotal evidence suggests that there have been varying untoward experiences with the vaccine that need to be investigated. In this study, we determined the prevalence, profiles, and predictors of Oxford/AstraZeneca vaccine side-effects among the vaccine recipients in the Tororo district of Eastern Uganda.

## 2. Materials and Methods

### 2.1. Study Design

This was a cross-sectional analytical study, on prevalence and predictors of Oxford/AstraZeneca vaccine side-effects among vaccine recipients in Tororo district community members, using quantitative methods. The population-based survey used secondary data extracted from COVID-19 vaccination registers and a telephone questionnaire interview.

### 2.2. Study Setting

The study was conducted in the Tororo district of Eastern Uganda, targeting all five vaccination sites allocated COVID-19 vaccination materials, for the initial phase of vaccination in Tororo district. Tororo district comprises 17 rural sub-counties, 2 town councils, and 2 municipal divisions. The district has a population of 597,500, with 291,300 males and 306,200 females [9]. The COVID-19 vaccination sites included: Tororo General Hospital, the three Health Center (HC) IVs (Mukuju, Mulanda, and Nagongera), and Osukuru HCIII. Uganda’s primary health care system is structured along the local government setup. The HCIII is a sub-county facility that offers all the services offered in HCII, plus maternity services, including facility delivery of expectant mothers. The staffing includes a midwife, a health assistant, and a records person. It is headed by a senior clinical officer, assisted by a clinical officer. The HCIV is a county facility that offers similar services as HCIII services and also serves as a referral facility for the lower units in the county, as well as providing emergency obstetric care, including cesarean sections. It is headed by a senior medical officer, assisted by a medical officer, a senior nursing officer, and a midwife. Allocation was guided by the initial registration of health workers and the available teachers, per the district registry records at the District Education Office. The vaccination exercise was under the direct supervision of immunization focal persons that mobilized and supervised teams of nurses, midwives, and data clerks, who put the vaccination and data entry into the registers. The health officer in charge of these facilities was responsible for doing overall supervision of the exercise, among other programs. The exercise was initially facility-based, but later involved targeted outreaches for organized entities, such as schools and factories.

### 2.3. Study Population

All the COVID-19 vaccine adult recipients, as listed in the COVID-19 vaccination registers of the five participating sites, with complete information in Tororo district, were eligible for this study. We excluded vaccine acceptors with hearing impairment, those without functional telephone contacts, those unable to sustain a telephone interview, and those that did not provide informed consent.

### 2.4. Sample Size

We enrolled all eligible participants, totaling 2204 between October 2021 and January 2022. This yielded a high (0.6% to 2.1%) absolute precision (half width of the 95% confidence interval) for side-effect prevalence estimates ranging from 2% to 50%.

### 2.5. Data Extraction

Personal profile data were derived from a review of the COVID-19 vaccination registers as secondary data sources, namely participants’ ID, client name, sex, and telephone contact, except for additional information that was included in the questionnaire. We used a data extraction tool to generate this data, as it was already collected in the COVID-19 vaccine registers.

### 2.6. Data Collection Methods

The cross-sectional observational study used telephone interviews for data collection. Training of research assistants ensured that they were familiar with the tool and could use it to collect quality data. Trained research assistants conducted questionnaire interviews with those with functional telephone contacts. They gathered data on side effects, related information (such as the previous infection with COVID-19), and other additional sociodemographic data. All the 2204 participants included in this study had functional telephone contacts. For those who died, we interviewed their caregivers or next of kin, as they were the ones who picked up the calls.

### 2.7. Measurement of Variables

The outcome variable was a side effect, following vaccination with AstraZeneca. It was defined as any untoward feeling experienced by a person, after being vaccinated. A question was asked, “Did you experience any of these listed side effects or untoward feelings after receiving the AstraZeneca vaccine?” The list included local side-effects, defined as: pain at the injection site, redness, swelling of lymph nodes, and local swelling; systemic side effects, defined as: tiredness, headache, nausea, diarrhea, vomiting, breathlessness, fainting, fever, muscle pains, joint pains, and chills; allergic side-effects, defined as: rash, skin burning, and red welts on face and lips; and other side effects.

The exposure variables included potential factors associated with Oxford/AstraZeneca vaccine side-effects. These were adopted from literature and included sociodemographic factors such as age, sex, education, religion, marital status, occupation, and residence. Others were individual participant characteristics such as vaccination status, previous infection with COVID-19, comorbidities, and healthcare-seeking behavior.

### 2.8. Statistical Analysis

First, we conducted exploratory data analyses to check the cleanliness of the data. We then summarized categorical data as proportions and continuous data using measures of central tendency [mean (SD), median (IQR)]. We conducted multivariable logistic regression to determine the factors associated with Oxford/AstraZeneca vaccine side-effects. Factors known to be associated with Oxford/AstraZeneca vaccine side-effects from the literature (for example, age, sex), and factors from the bi-variable analysis with a ***p***-value less than 0.2 (occupation, education level, marital status) were included in the multivariable analysis. We used Stata version 15.0 for analysis.

## 3. Results

### 3.1. Participant Characteristics

A total of 2204 participants were recruited, of whom 68.7% (1515/2204) were less than 50 years old. The majority, 59.4% (1310/2204), were male. In terms of education, more than half, 57.4% (1264/2204), had tertiary education. Concerning religion, the majority (88.7%) were Christians. Most of the participants, 86.2% (1900/2204), were married. In terms of occupation, the majority of the population were teachers, 23.2% (512/2203), followed by health workers at 17.1% (377/2203), and security at 8.7% (191/2203). Three-point four percent (75/2204) of the population had been infected with COVID-19 before the vaccination exercise. The details are in Table 1 below.

### 3.2. Prevalence of Side-Effects to Oxford/AstraZeneca Vaccine in Tororo District

A total of 603 out of 2204 experienced side-effects, representing (27.4%) (Figure 1). Of those who experienced side-effects, 253/603 (42.0%) experienced local side-effects, 449/603 (74.5%) experienced systemic side-effects, 13/603 (2.2%) had allergic reactions, and 171/603 (28.4%) reported other side-effects. Other side-effects included loss of libido, changes in appetite, body weakness, cough, and flu.

Only 424/603 participants had responses to when side-effects occurred. Of these, 268/424 (63.2%) of the participants experienced side-effects after the first dose, 44/424 (10.4%) experienced side-effects after the second dose, and 112/424 (26.4%) experienced side-effects after both doses. Ten participants did not receive the second dose of the vaccine because of the side-effects they experienced after the first dose. Among those that experienced side effects after both doses of the vaccine, 74/112 (66.1%) were reported to have been affected more in terms of the severity of symptoms by the first dose.

### 3.3. Side-Effect Profile

For local side-effects, the majority 247/253 (97.6%) reported pain at the injection site, 16/253 (6.3%) swelling at the injection site, 3/253 (1.2%), swollen armpit lymph nodes, and 1/253 (0.4%) had redness at the injection site. For systemic side-effects, most, 218/449 (48.5%), experienced headache, 203/449 (45.2%) had tiredness, and 134/449 (29.8%) reported fever. Other side-effects are shown in Figure 1 and Figure 2 below:

### 3.4. Health Care Seeking following Oxford/AstraZeneca Vaccine Side-Effects

Almost half of the participants, 269/424 (63.4%) did not seek any health care after experiencing side-effects. However, a total of 64/424 (15.1%) of the participants sought medical attention in a health facility. These probably represented those with serious-side effects, as a visit to a health facility may be a proxy indicator for a serious medical condition. Among those who visited a health facility, 34 out of 64 visited a private facility (Table 2).

### 3.5. Medications following Oxford/AstraZeneca Side-Effects

A total of 156 respondents reported the use of medications following Oxford/AstraZeneca side-effects. Most of the respondents, 96/156 (61.5%), used paracetamol following COVID-19 vaccination side-effects. Other medications used are in Table 3.

### 3.6. Deaths following Oxford/AstraZeneca Vaccination

Of the participants, 7/424 died after COVID-19 side-effects (Table 4). The causes of death were probably not directly related to the side-effects as shown in the table below. Information about the death was obtained via telephone interview. However, it was not possible to infer an association between death and the side-effect of Oxford/AstraZeneca.

### 3.7. Factors Associated with Experiencing Side-Effects to Oxford/AstraZeneca Vaccine

Previous infection with COVID-19 (adjusted odds ratio (AOR): 4.3, 95% confidence interval (95% CI): 2.7–7.0, *p* < 0.001) and being female (AOR: 1.3, 95% CI: 1.1–1.6, *p* = 0.004) were positively associated with experiencing side-effects of the AstraZeneca COVID-19 vaccine. Being a security officer (AOR: 0.4, 95% CI: 0.2–0.6, *p* < 0.001) was negatively associated with experiencing side-effects of the COVID-19 AstraZeneca vaccine. Participants previously infected with COVID-19 were 4.3 times more likely to experience Oxford/AstraZeneca vaccine side-effects than those who weren’t. Females were 1.3 times more likely to experience side-effects than males. Security officers were 0.4 times less likely to experience side-effects of the Oxford/AstraZeneca vaccine compared to participants of different occupations. Table 5 shows the factors associated with experiencing side-effects to Oxford/AstraZeneca vaccine.

## 4. Discussion

In this study, we investigated the side-effects and associated factors following COVID-19 vaccination with the Oxford/AstraZeneca vaccine among priority populations comprising health workers, teachers, security personnel, the elderly above fifty, and all adults between 18 and 50 with underlying conditions. This was a COVID-19 vaccine-naïve population, as they were the first beneficiaries of this service in the phased approach the government undertook to vaccinate its eligible citizens. In our study, 27.4% of the participants experienced one or more side-effects. Studies conducted elsewhere reported higher findings, as compared to our findings. For instance, a study carried out in Saudi Arabia reported a 68.5% prevalence of side-effects [10]. Another study carried out in Jordan on a vaccine naïve population but comparing AstraZeneca with Pfizer and Sino pharm, reported an 89.9% prevalence of side-effects [11]. The same study revealed that more side-effects were significantly associated with the Oxford/AstraZeneca vaccine than other vaccines. Additionally, a study conducted among health workers in Ethiopia reported a 91.3% prevalence of AstraZeneca COVID-19 vaccine side-effects [12]. The differences could be explained by differences in populations, as some of them could present differing thresholds for discomfort, as well as the nocebo effect in that, some populations could be more averse to rumors and misinformation. The nocebo effect can modulate the outcome of a given therapy negatively [13]. In this case, it could be induction or worsening of side-effects of the Oxford/AstraZeneca vaccine.

We found that being female was positively linked to experiencing side-effects. This is contrary to a study conducted in Ethiopia on health workers that found no association [14]. However, similarly, a cross-sectional survey among recipients of the COVID-19 vaccine in the general population in Saudi Arabia reported a higher prevalence of side-effects among women than men, after either dose [15]. We cannot rule out the nocebo effect in explaining gender relations, as some people could have experienced side-effects out of expectation. This is emphasized in a report by Winfried Rief of the JAMA health forum, that states that the very fear of side effects can amplify or induce side-effects [16].

The strong relation between side-effects with previous COVID-19 infection could be a result of a primed body with natural immunity, developing from a previous infection reacting more aggressively to the vaccine [17]. Similarly, a prospective observational study conducted in the United Kingdom showed such a strong linkage between a previous COVID-19 infection and side-effect experience, a 1.6 times higher likelihood of side-effects in those with the previous infection [4]. A significant association between side-effects following the previous infection was quite apparent, suggesting the possibility of the vaccine landing on a primed immune system that probably overreacted [17].

A small number (15.1%) of the participants who had side-effects sought medical care from a health facility. We can take these as those with serious side-effects, as a visit to the health facility could be a proxy indicator for a serious condition in our setting. This was way above the 2% serious side-effects reported by a study in the United States [18]. The difference could be accounted for by a difference in the definition of serious side-effects. However, a study conducted in England concluded that there aren’t enough data to conclude serious adverse events following COVID-19 vaccination, as not enough clinical trials and long-term follow-up have been performed [19].

Among those that experienced side-effects after both doses of the vaccine, 76/112 (67.9%) were reported to have been affected more by the first dose. Similar results were reported by a study in Poland of participants being affected more by the first dose of the vaccine [20].

Security personnel had a statistically significant protective relationship. Perhaps the hardened nature due to their training and work makes them less likely to report minor events as side-effects. However, these are generally fit people but also “macho” in nature by training, and possibly are more inclined towards not reporting minor events.

Our study provides evidence of mild symptoms, as most of the side-effects were managed conservatively (did nothing), followed by self-medication, using mainly paracetamol tablets. This is in agreement with a study conducted in Ethiopia amongst health workers that reported that 64% of the people who experienced side-effects used paracetamol as a remedy [14]. Up to 18/424 (4.2%) of those who experienced side-effects had ongoing symptoms at the time of the study, on average six months later. Seven participants died from causes that may not necessarily have been related to the vaccine; two from accidents, one from TB, malaria, stroke, hypertension, and diabetes mellitus, respectively, on average 3 months post-vaccination. Ten participants couldn’t proceed to receive the second dose because of the side-effects.

### Study Limitations

This population-based survey used telephone interviews to gather quantitative data using a structured questionnaire. We proposed to reach an estimated 7834 people who had received the Oxford/AstraZeneca vaccine from the first 8000 doses released to Tororo as of July 2021. Of the over 7834 targeted population, only 5750 were deemed to have complete data to be contacted, and of these, only 2204 were reached. Some telephone contacts were not available or unreachable. Some contacts were duplicated in the register. This could have also resulted in information bias, as the participants who could not be accessed could have reported side-effects.

The retrospective nature of our study paused a risk of recall limitations, as this study took place more than six months after the vaccination exercise. Furthermore, COVID-19 considerations have impacted some practical aspects of this study.

## 5. Conclusions

Following vaccination with the Oxford/AstraZeneca vaccine, participants reported side-effects that were majorly local and systemic. Most of the side-effects were mild and self-limiting. Being female and having had a previous COVID-19 infection were positively linked to experiencing side-effects after vaccination with the Oxford/AstraZeneca vaccine, while being security personnel was negatively associated with experiencing side-effects after vaccination with the Oxford/AstraZeneca vaccine. We recommend campaigns to disseminate correct information about potential side-effects of the Oxford/AstraZeneca vaccine, and to strengthen surveillance for adverse events following vaccination.

## Figures and Tables

**Figure 1 ijerph-19-15303-f001:**
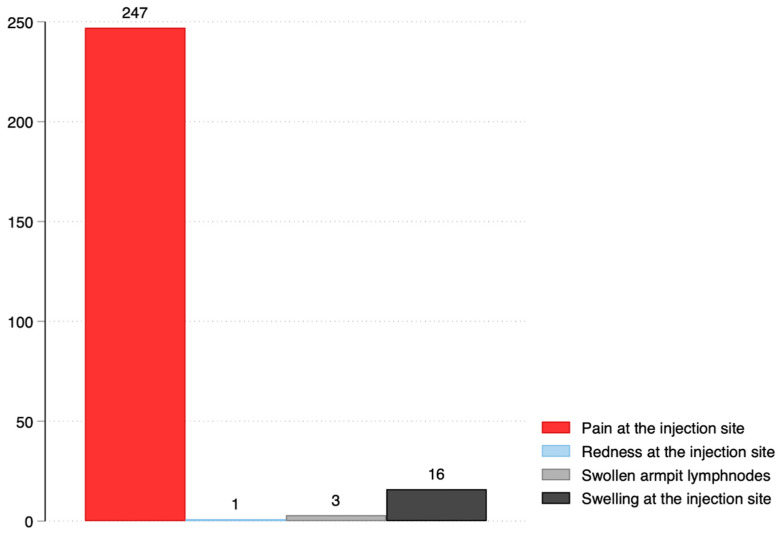
Local side-effects following vaccination with Oxford/AstraZeneca.

**Figure 2 ijerph-19-15303-f002:**
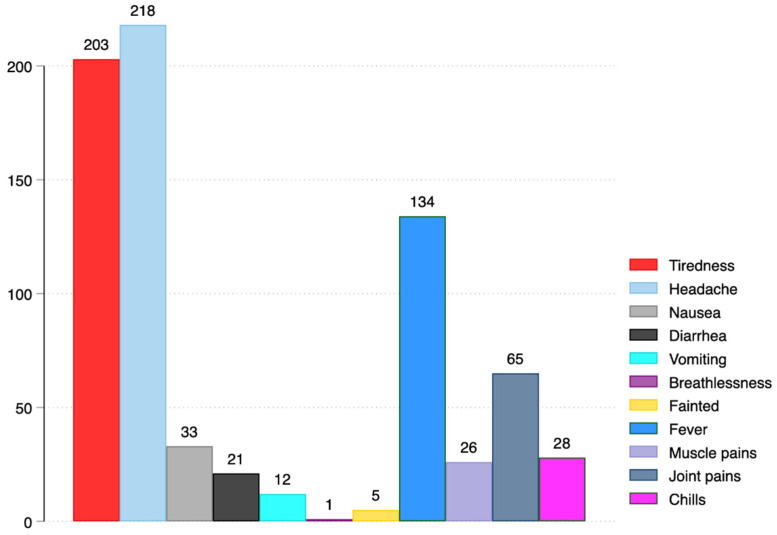
Systemic side-effects following vaccination with Oxford/AstraZeneca.

**Table 1 ijerph-19-15303-t001:** Characteristics of study participants.

Characteristic, *n* = 2204	Frequency (*n*)	Percentage
**Age**		
<50	1515	68.7
>50	689	31.3
**Sex**		
Male	1310	59.4
Female	894	40.6
**Education level (*n* = 2203)**		
Primary	398	18.1
Secondary	541	24.5
Tertiary	1264	57.4
**Religion (*n* = 2203)**		
Christian	1953	88.7
Moslem	206	9.3
Hindu	44	2.0
**Marital status**		
Single	304	13.8
Married	1900	86.2
**Occupation (*n* = 2203)**		
Teacher	512	23.2
Health worker	377	17.1
Security	191	8.7
Others *	1123	51.0
**Previous COVID–19 Infection**		
Yes	75	3.4
No	2129	96.6

* Others included: business people, farmers, drivers, students.

**Table 2 ijerph-19-15303-t002:** Health care seeking following AstraZeneca side-effects.

Place of Seeking Care (*n* = 424)	Frequency (*n*)	Percentage (%)
Did not seek health care	269	63.4
Consulted CHW	21	5.0
Consulted traditional healer	3	0.7
Visited health center	18	4.3
Visited hospital	12	2.8
Visited private clinic	34	8.0
Self-medication	72	17.0
Other *	2	0.5

* other used home remedies, taking juice and a lot of fluids.

**Table 3 ijerph-19-15303-t003:** Medications following Oxford/AstraZeneca vaccine side-effects.

Medication (*n* = 156)	Frequency (*n*)	Percentage (%)
Herbs	5	3.2
Paracetamol	96	61.5
Diclofenac	33	21.2
Amoxicillin	11	7.1
Azithromycin	3	1.9
Chloroquine	2	1.3
Ciprofloxacin	6	3.9
Vitamin C	4	2.6
Dexamethasone	13	8.3
Others *	10	6.4
Don’t know	18	11.5

* others included: metronidazole, prednisolone, artesunate, artemether–lumefantrine, fluids.

**Table 4 ijerph-19-15303-t004:** Deaths following receipt of Oxford/AstraZeneca vaccine.

Participant	Cause of Death	Period from the Date of the Second Dose of the Vaccine
**1**	Diabetes complication	Three weeks
**2**	Hypertension/stroke	Three weeks
**3**	Accident	Two weeks
**4**	Malaria	One month
**5**	Sudden death	Three months
**6**	Accident	Five months
**7**	Tuberculosis	Two months

**Table 5 ijerph-19-15303-t005:** Factors associated with experiencing side-effects of Oxford/AstraZeneca vaccine.

Characteristic	Had Side Effects	COR	95% CI	*p*-Value	AOR	95% CI	*p*-Value
**Age**							
<50	415 (68.8)	1			1		
≥50	188 (31.2)	1.0	0.8–1.2	0.958	1.0	0.8–1.2	0.942
**Sex**							
Male	322 (53.4)	1			1		
Female	281 (46.6)	1.4	1.2–1.7	<0.001	1.3	1.1–1.6	**0.004 ***
**Marital status**							
Single	85 (14.1)	1			1		
Married	518 (85.9)	1.0	0.7–1.3	0.800	0.9	0.7–1.2	0.595
**Previous COVID-19 infection**							
No	557 (92.4)	1			1		
Yes	46 (7.6)	4.5	2.8–7.2	<0.001	4.3	2.7–7.0	**<0.001 ***
**Education level**							
Primary	114 (18.9)						
Secondary	160 (26.5)	1.0	0.8–1.4	0.756	1.1	0.8–1.5	0.423
Tertiary	329 (54.6)	0.9	0.7–1.1	0.304	0.8	0.6–1.1	0.163
**Occupation**							
Teacher	151 (25.04)	1			1		
Health worker	110 (18.2)	0.9	0.7–1.2	0.549	0.9	0.7–1.2	0.561
Security	27 (4.5)	0.4	0.3–0.6	<0.001	0.4	0.2–0.6	**<0.001 ***
Others	315 (52.2)	1.0	0.7–1.3	0.919	0.8	0.6–1.1	0.152

COR—crude odds ratio; CI—confidence interval; AOR—adjusted odds ratio; * statistically significant at a *p*-value < 0.05.

## Data Availability

The datasets used and/or analyzed during the current study are available from the corresponding author.

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
