# Peer review of "Side-Effects following Oxford/AstraZeneca COVID-19 Vaccine in Tororo District, Eastern Uganda: A Cross-Sectional Study"

_ijerph, 2022, doi:10.3390/ijerph192215303_

Round 1

Reviewer 1 Report

The manuscript describes the  results of phone in interviews regarding to side effects of Oxfort/AstraZeneca vaccine in Eastern Uganda region. The results were presented very clearly and the manuscript was written well. Considering the resistance to get vaccinated in the world, this study bring a light to the issue by determining the factors contributes to this opposition.

Some minor revision suggestions as follows:

1) Abstract: Lines 20-22. Suggest revisions the sentence  "......of the participants experienced one or more side effects as follows: local side effects, systemic side effects, allergic side effects, and other side effects."  If the authors give numbers and percentages, when the reader adds up that numbers and percentages, it is more than 603 and 100% because some people experienced more than one side effect. This way the numbers are not causing any confusion.

2) Results: Line154-155: suggest to change the sentence to "....were less than 50 years old."

3)Results: Line 159: The sentence should not start with a number. it should be "Three point four percent ..."

4) Results:  Figure 1: there is no need for figure 1 since the percentage is stated in the text.

5) Results: Table 2 and Figure 2: Both of these shows the same data therefore there is no need to repeat. Please use only table or only graph. If you decide to use graph, I suggest to add numbers to top of column for each column.

6) Results: Figure 3 and Table 3: Again both represents the same data . Please use only one of them and again if you use graph, please add numbers to top of each column.

7) Results: Table 4 and Figure 4: Same comment as comments 5 and 6.Only one of them will be sufficient.

8) Discussion: Line 250: Please add "one or more" front of "side effects".

9) Discussion: Line 302: Please change "Hypertension" to "hypertension". No need to capitalize "h".

10) Discussion: Line 303: Last sentence states that 6 participants did not get 2nd dose of vaccine because of side effects but in the abstract (line23) it was stated that the number was 10. Which one is correct? 6 or 10?

11) Discussion: Line 312. Please delete extra space from of "This" .

Author Response

1) Abstract: Lines 20-22. Suggest revisions the sentence  "......of the participants experienced one or more side effects as follows: local side effects, systemic side effects, allergic side effects, and other side effects."  If the authors give numbers and percentages, when the reader adds up that numbers and percentages, it is more than 603 and 100% because some people experienced more than one side effect. This way the numbers are not causing any confusion.

Thank you for the suggestion we have revised the sentence to reflect this idea.

2) Results: Line154-155: suggest to change the sentence to "....were less than 50 years old."

Thank you for the suggestion, this has been changed.

3)Results: Line 159: The sentence should not start with a number. it should be "Three point four percent ..."

This has been rectified

4) Results:  Figure 1: there is no need for figure 1 since the percentage is stated in the text.

Thank you, we have removed figure 1

5) Results: Table 2 and Figure 2: Both of these shows the same data therefore there is no need to repeat. Please use only table or only graph. If you decide to use graph, I suggest to add numbers to top of column for each column.

Thank you, this has been rectified. We decided to use the graphs and added numbers on top of each column

7) Results: Table 4 and Figure 4: Same comment as comments 5 and 6.Only one of them will be sufficient.

This has been revised

 8) Discussion: Line 250: Please add "one or more" front of "side effects".

Thank you, This has been rectified

9) Discussion: Line 302: Please change "Hypertension" to "hypertension". No need to capitalize "h".

This has been revised

10) Discussion: Line 303: Last sentence states that 6 participants did not get 2nd dose of vaccine because of side effects but in the abstract (line23) it was stated that the number was 10. Which one is correct? 6 or 10?

Thank you for the observation. This has been corrected accordingly

11) Discussion: Line 312. Please delete extra space from of "This" .

This has been rectified

Reviewer 2 Report

The manuscript of Jagire Onyango and colleagues investigated the prevalence, profiles, and predictors of Oxford/AstraZeneca vaccine side effects among the vaccine recipients in Tororo district in Eastern Uganda by phone interview. The method of this study induced the risks of acquiring bias/inaccurate data from patients. Although some of these limitations have been documented in the limitation section, it is a big defect of the study. 

Major comments:

1.     The authors found being female was positively linked to experiencing side effects, however, they cannot rule out the nocebo effect in explaining gender relations. In addition, line 291-293 “Security personnel had a statistically significant protective relationship. Perhaps the hardened nature due to their training and work makes them less likely to report minor events as side effects.” Two out of three significant associated factors with side effects are not convinced. Is this method appropriate to evaluate side effects for this study?

2.     The authors analysis factors associated with experiencing side effects including ages, sex, marital status and others. Factors related to health status should be involved, such as if they have other diseases.

Minor comments:

1.     Please modify the figure colors to be red/green color blind friendly.

2.     Please describe the time of study performed at the beginning of the manuscript.

Author Response

1.     The authors found being female was positively linked to experiencing side effects, however, they cannot rule out the nocebo effect in explaining gender relations. In addition, line 291-293 “Security personnel had a statistically significant protective relationship. Perhaps the hardened nature due to their training and work makes them less likely to report minor events as side effects.” Two out of three significant associated factors with side effects are not convinced. Is this method appropriate to evaluate side effects for this study?

We believe our methods were robust. We discuss the potential limitations in our study. The only way to rule out a nocebo effect was having a group that received a placebo. That way, the total side effects attributable to the vaccines would be calculated. However, it would not be ethical to give one group a nocebo and yet the efficacy of vaccines had already been determined.

We also found that previous vaccination was associated with higher odds of experiencing a side effect, this is line with biological plausibility.

2.     The authors analysis factors associated with experiencing side effects including ages, sex, marital status and others. Factors related to health status should be involved, such as if they have other diseases.

Thank you for this comment Unfortunately, we did not collect data on participants health status

1.     Please modify the figure colors to be red/green color blind friendly.

Thank you, they have been modified.

2.     Please describe the time of study performed at the beginning of the manuscript.

The study was conducted between October 2021 and January 2022. We have included this in the abstract and methods section.

Round 2

Reviewer 2 Report

The authors have improved the manuscript. Figure colors need to be modify for color blind readers (should avoid green and red in the same figure).

Author Response

Reviewer’s comment

Response to comment

The authors have improved the manuscript. Figure colors need to be modify for color blind readers (should avoid green and red in the same figure).

To ensure that we used color-blind friendly graphs; we used the cleanplots scheme in Stata:

https://www.trentonmize.com/software/cleanplots

This scheme ensures that colors chosen are distinguishable when printed in black and white and that they are friendly to color blind people. Further more, we used the website below to check the colors that we have included for their representations in a color blinded people with protanopia, deueranopia and tritanopia:

https://davidmathlogic.com/colorblind/#%23C80000-%2382C0DF-%23808080-%23000000-%2305CECE-%23881161

We have also intentionally avoided having red and green in the same graph.

We believe the graphs we have chosen are friendly to people with color blindness.